# Interpretable Convolutional Neural Networks for Preterm Birth Classification

**Irina Grigorescu, Lucilio Cordero-Grande, A David Edwards, Jo Hajnal, Marc Modat, Maria Deprez**
*School of Biomedical Engineering & Imaging Sciences, King's College London, London, UK*

## Abstract

The use of convolutional neural networks (CNNs) for classification tasks has become dominant in various medical imaging applications. At the same time, recent advances in interpretable machine learning techniques have shown great potential in explaining classifiers' decisions. Layer-wise relevance propagation (LRP) has been introduced as one of these novel methods that aim to provide visual interpretation for the network's decisions. In this work we propose the application of 3D CNNs with LRP for the first time for neonatal $T_2$-weighted magnetic resonance imaging (MRI) data analysis. Through LRP, the decisions of our trained classifier are transformed into heatmaps indicating each voxel's relevance for the outcome of the decision. Our resulting LRP heatmaps reveal anatomically plausible features in distinguishing preterm neonates from term ones.

**Keywords:** preterm birth, classification, layer-wise relevance propagation

## 1. Introduction

In medical imaging, interpretation of deep neural networks is a current topic of interest as it promises a way towards understanding and validating machine learning predictions. In classification problems, layer-wise relevance propagation (Bach et al., 2015a) has been introduced as a novel method of illustrating network decisions. More specifically, LRP assigns a 'relevance' quantity to every voxel in the input image, where a high value represents a strong influence of that particular voxel towards the network's decision. This not only provides a means of classifier interpretability, thus increasing the medical experts' trust, but can also help with patient specific analysis.

In this project, we aim to assess the potential viability of the LRP method to uncover underlying features of preterm birth. For this, we propose a deep learning framework that is trained to classify $T_2$w MRI volumes of neonates into either *term* (born after 37 weeks post-menstrual age) or *preterm* (born before 37 weeks post-menstrual age) babies. Then, we apply the LRP method to identify the relevant features of the input images and we produce group specific relevance maps.

## 2. Methods

**Data preprocessing.** In this work we use $T_2$w 3D magnetic resonance imaging volumes that were acquired as part of the developing Human Connectome Project (dHCP[1]). More

---

1. http://www.developingconnectome.org/

specifically, we use a dataset of 157 MRI scans of infants born between $23 - 42$ weeks gestational age (GA) and scanned at term-equivalent age (after 37 weeks GA). To aid with our interpretability part of this study, we perform a series of pre-processing steps to our dataset. First, we register all of our data to a common 40 weeks gestational age atlas space using non-rigid B-spline registration (B-spline control point spacing 10mm) available in the IRTK (Rueckert et al., 1999) software toolbox and we downsample the images to $1mm^3$ isotropic resolution. This was done in order to remove anatomical differences and to focus on intensity differences, thus allowing for population comparison of the most relevant voxels for the classification task. Finally, we perform skull-stripping and we crop the volumes to a $128 \times 96 \times 96$ size in order to allow an entire 3D volume to be fed into the network at one particular time.

**Classifier network.** The proposed 3D convolutional neural network (3D-CNN) architecture uses $T_2$w volumes of neonates at term equivalent age as inputs and classifies them into either *preterm* or *term*. A schematic illustration of the overall network is shown in Figure 1. The network contains repeated blocks of $3 \times 3 \times 3$ convolutions (with a stride of 1), batch normalization (Ioffe and Szegedy, 2015), rectified linear unit (ReLU) activations and $2 \times 2 \times 2$ average pooling layers (with a stride of 1), followed by two fully connected layers. The network outputs the probabilities of an input image belonging to either of the two classes.

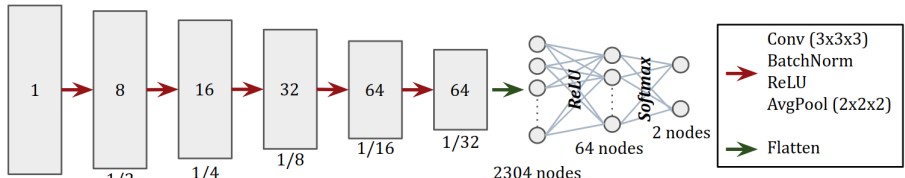

Figure 1: The proposed network architecture for our classification task. Each rectangle represents a 3D volume, where the number of channels is shown inside the rectangle, while the spatial resolution with respect to the input volume is shown underneath.

The network was trained by minimizing a categorical cross entropy loss function using the Adam optimizer with the default parameters ($\beta_1 = 0.9$ and $\beta_2 = 0.999$). The learning rate was varied in a decaying cyclical fashion (Smith, 2015) with a base learning rate of $10^{-5}$ and a maximum learning rate of $10^{-3}$. The data was split into 90% training and 10% testing sets. We performed a 10-fold cross-validation on the training set to find the set of hyperparameters that maximize our model's generalization performance. To account for the class imbalance between the *preterm* and *term* classes, we introduced a stronger weight in the loss function for the under-represented class (*preterm*).

**Relevance maps.** Layer-wise relevance propagation (Bach et al., 2015a) is a backward propagation technique that was found to be applicable in a variety of computer vision applications (Arbabzadah et al., 2016) (Bach et al., 2015b) and medical data (Sturm et al., 2016). This method assigns a 'relevance' score to each input voxel by iteratively propagating through the network each layer's output to its predecessors until the input layer is reached (Montavon et al., 2018). This redistribution rule is guided by a conservation principle, in which every neuron in the architecture receives a share of the network output (Bach et al., 2015a). LRP is closely related to deep Taylor decomposition (Montavon et al., 2015) and is

applicable to a wide range of network architectures. In this work we implemented the LRP rules, as described in (Montavon et al., 2018), for our 3D classification network and applied them to compute relevance maps for both *preterm* and *term* neonates.

## 3. Results

Our network obtained a 94% accuracy score, with a true positive rate of 100%, and a true negative rate of 86% on the test set. We investigated the misclassified neonates and our results show that they were born closer to the 37 weeks threshold than all the other preterm babies. Next, we computed the LRP maps using the $\alpha\beta-$ rules for all the correctly classified images, where $\alpha - \beta = 1$ and $\beta \geq 0$ (Bach et al., 2015b). Figure 2 shows our resulting average relevance maps for the two different classes. Our results show that the most prominent feature was the cerebrospinal fluid (CSF), in agreement with previous clinical literature where it was found that preterm babies have more CSF and less cortical folding due to impaired brain growth (Alexandrou et al., 2014).

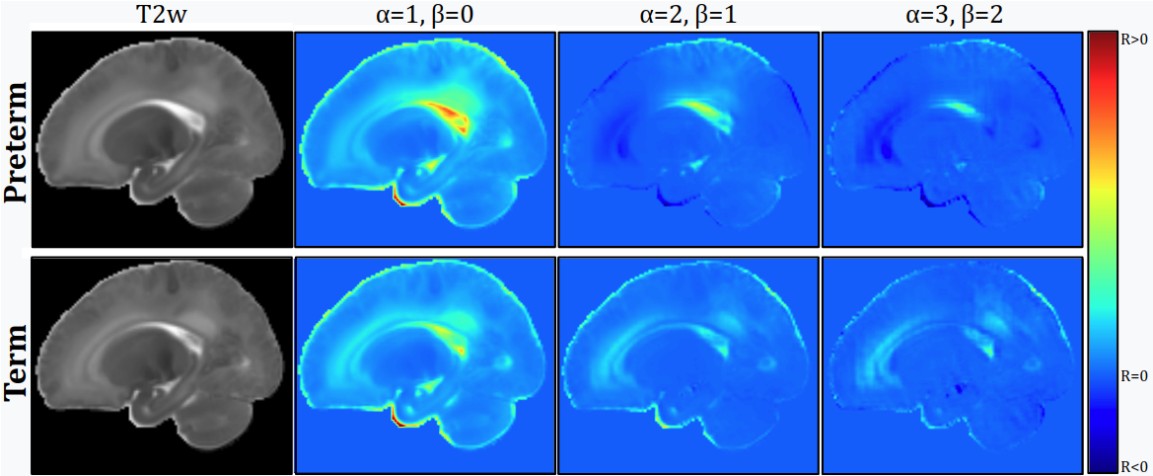

Figure 2: Average brain images for both *preterm* and *term* infants together with their corresponding relevance maps with varying $\alpha$ and $\beta$ parameters.

## 4. Discussion and Future Work

In this study we showed the application of 3D convolutional neural networks with layer-wise relevance propagation for the first time for neonate $T_2$w magnetic resonance imaging. Our preliminary analysis showed that CSF is an important feature for distinguishing between *term* and *preterm* birth. For future work we plan to investigate the differences in shape as well as intensity in the population data. Moreover, we aim to include different modalities, such as diffusion MRI, in our proposed framework and to explore and compare our current method with different interpretability techniques.

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
