# OpenReview forum: "Interpretable Convolutional Neural Networks for Preterm Birth Classification"
_MIDL.io/2019/Conference/Abstract — MIDL Abstract 2019_

### Official Review · AnonReviewer2 · 2019-04-29
**Unclear novelty and inconclusive results**

**Rating:** 2
**Confidence:** 2

**Review:**

The authors proposed a method to improve the interpretability of neural network predictions applied on medical images. The method is described using 3D CNN for binary classification and then layer-wise relevance propagation (LRP) to reveal interpretable features in MRI scan of infants brains by the network. The method is able to identify critical characteristics to the network's decision of preterm birth.
The paper may have the following weaknesses:
1) Unclear novelty, although it is novel to apply this method on medical images, LRP was previously used on MNIST and PASCAL VOC, and this work can be seen as a straightforward application if the original LRP is not improved with modifications. Therefore the novelty of the work is unclear.
2) Quantitative results, the authors visualised the relevance maps for scans of both term and preterm  babies, although differences can be seen by visual inspection in CSF and cortical folding, it's inconclusive without quantitative analysis statistically confirming the differences and excluding the difference comes from individual diversity. In line 5 of the result section, an average relevance map is used, how is this map calculated?
Also, the author may also need to compare the described method with other related work in order to position the work in its research stream and highlight its advantages.

---

### Official Review · AnonReviewer1 · 2019-04-30
**novel, interesting study**

**Rating:** 4
**Confidence:** 2

**Review:**

* Methods are clearly described, and results appear promising. Novel idea to apply layer-wise relevance propagation to the problem of preterm birth classification, and the features emerging as 'relevant' are reported to be plausible based on prior knowledge.
* For context, it would be helpful to know how the authors' classification results compare to any prior results in the literature addressing a similar question (if any exist)

---

### Decision · Program_Chairs · 2019-05-06
**Acceptance Decision**

Accept